# WALT: WEB AGENTS THAT LEARN TOOLS

**Viraj Prabhu, Yutong Dai, Matthew Fernandez, Krithika Ramakrishnan, Jing Gu, Yanqi Luo, Silvio Savarese, Caiming Xiong, Junnan Li, Zeyuan Chen, Ran Xu**
Salesforce Research
{viraj.prabhu, ran.xu}@salesforce.com

## ABSTRACT

Web agents promise to automate complex browser tasks, but current methods remain brittle—relying on step-by-step UI interactions and heavy LLM reasoning that break under dynamic layouts and long horizons. Humans, by contrast, exploit website-provided functionality through high-level operations like search, filter, and sort. We introduce WALT (Web Agents that Learn Tools), a framework that reverse-engineers latent website functionality into reusable invocable tools. Rather than hypothesizing ad-hoc skills, WALT exposes robust implementations of automations already designed into websites—spanning discovery (search, filter, sort), communication (post, comment, upvote), and content management (create, edit, delete). Tools abstract away low-level execution: instead of reasoning about *how* to click and type, agents simply call `search(query)` or `create(listing)`. This shifts the computational burden from fragile step-by-step reasoning to reliable tool invocation. On VisualWebArena and WebArena, WALT achieves state-of-the-art success rates (52.9% on VisualWebArena, 50.1% on WebArena) with fewer steps and less LLM-dependent reasoning. On Online-Mind2Web, a benchmark of 139 real-world websites, WALT autonomously discovers 252 tools and improves success rate by 20.5% over a tool-free baseline, establishing a robust and generalizable paradigm for browser automation.

**Code:** `https://github.com/SalesforceAIResearch/WALT`

## 1 INTRODUCTION

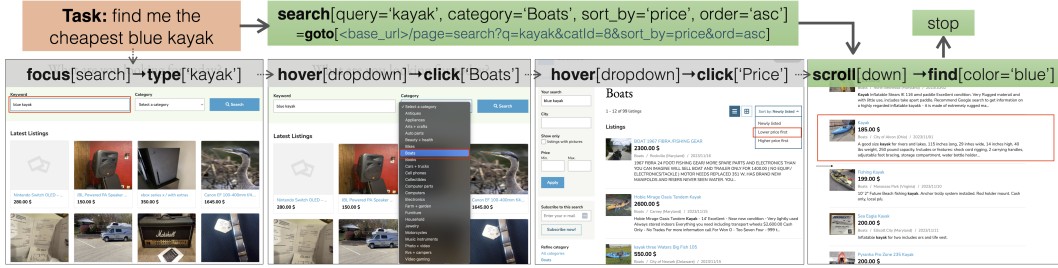

Figure 1: **WALT transforms browser agent automation from brittle step-by-step reasoning to efficient tool-based abstraction.** For the task "find the cheapest blue kayak," traditional web agents execute a lengthy sequence of primitive UI actions focusing on search boxes, hovering over drop-downs, clicking categories, and sorting and scanning results. Our method WALT (Web Agents that Learn Tools), designs a deterministic tool that exposes this website-provided functionality to the agent: `search(query='blue kayak', category='Boats', sort_by='price')`, reducing execution from 8+ fragile UI steps to 1 robust operation.

Consider searching for the cheapest blue kayak on a classifieds page (Fig. 1): existing web agents reason through each step—how to interact with the search box, locate filter controls, determine the correct sort option—while simultaneously handling implementation details like element selection and timing. In contrast, humans naturally think about this task in terms of website functionality: "search for kayaks, filter by price, identify the first blue one." They abstract away the implementation details and focus on what they want to accomplish, not how the interface mechanics work.

This human capability stems from recognizing reusable patterns across websites — not only in search and filtering, but also in content creation and management (e.g., creating, editing, deleting listings) and social interactions (e.g., commenting, messaging, upvoting). Humans leverage this prior knowledge to quickly adapt their interaction strategies to new websites. In the web agent context, this intuition has inspired work on discovering "skills" (Wang et al., 2025b) for web agents—reusable action sequences that encapsulate common interaction patterns and can be applied across similar website elements or tasks.

However, existing skill discovery approaches suffer from two key limitations in *which* skills are discovered and *how* they are implemented. First, they either mine skills only from successful trajectories (Wang et al., 2025b; Sarch et al., 2024; Wang et al., 2025a)—codifying existing behaviors—or require agents to hypothesize useful automations (Zheng et al., 2025), often yielding unintuitive, overly specific, or irrelevant skills. Second, both approaches implement skills as brittle UI action sequences, highly sensitive to dynamic elements and design changes.

We propose WALT (Web Agents that Learn Tools). Unlike prior "skills" or "workflows," which are agent-induced action sequences, our tools correspond to *website-provided functionality*—search bars, filters, sorting mechanisms, commenting systems, and navigation controls—that site designers have already engineered as robust automations. Each tool is exposed to the agent as a high-level deterministic call, with an underlying implementation discovered and validated through reverse-engineering. This reframing shifts the agent's capability frontier: instead of learning brittle approximations of interaction patterns, WALT surfaces the functionality already embedded in websites as reliable, reusable tools.

On each website, WALT follows a demonstrate-generate-optimize-test loop for each identified tool: (1) a web agent comprehensively demonstrates the functionality (*e.g.*, all filters and sort options for search); (2) a tool generation agent maps execution traces to structured tools with validated input schemas, prioritizing deterministic actions but allowing agentic steps for dynamic elements; (3) the tool is optimized by replacing UI sequences with more robust URL manipulation through API reverse-engineering; (4) a test agent verifies functionality against pre-vetted test inputs.

This abstraction transforms the agent's computational burden: instead of reasoning about "how do I search for X, then filter by Y, then sort by Z" through complex UI sequences, the agent simply calls `search(X)`, `filter(Y)`, `sort(Z)` and focuses on higher-level planning. Tool discovery and optimization happen offline during website exploration, ensuring both efficiency and reliability.

We benchmark our method on VisualWebArena (Koh et al., 2024a) and WebArena (Zhou et al., 2024), discovering over 50 reusable tools spanning search and filtering, content creation and management (e.g., create, edit, delete listings), and communication or social interactions (e.g., commenting, messaging, upvoting). WALT achieves state-of-art success rates of 52.9% on VisualWebArena and 50.1% on WebArena, significantly outperforming prior work. We further validate real-world generalizability on Online-Mind2Web (Xue et al., 2025), a benchmark of 139 live websites, where WALT discovers 252 tools and improves success rate by 20.5% relative over a controlled tool-free baseline, achieving near-parity with the Claude Computer-Use Agent without any specialized training. Ablation studies further reveal that our proposed contributions — discovered tools, multimodal DOM parsing, and external verification – yield gains in both success rates (10%-30% across splits) and efficiency (1.3-1.4x fewer steps on average). Overall, WALT transforms browser agent automation from brittle step-by-step reasoning to efficient tool-based abstraction.

## 2 RELATED WORK

**Web Agents.** Agents capable of directly operating a browser to perform tasks hold promise for automating online tasks. Prior work advances web agents along four axes: **Perception** concerns what the agent sees and how it grounds elements: some methods parse raw HTML (Gur et al., 2022; Deng et al., 2023), others process full-page screenshots with vision-language models (Furuta et al., 2023; He et al., 2024), often augmented by Set-of-Mark (SoM) visual prompts (Yang et al., 2023); recent work improves page understanding and grounding via prompting (Zheng et al., 2024; Yao et al., 2023) and task-specific training (Furuta et al., 2023; Zhang et al., 2025a; Pahuja et al., 2025; Qi et al., 2024). **Planning** scales test-time exploration with search (e.g., MCTS and related variants) to choose better action sequences (Koh et al., 2024b; Putta et al., 2024; Yu et al., 2024; Gu et al., 2024). **Reasoning** enhances step selection through chain-of-thought and ReAct-style prompt-

ing (Wei et al., 2022; Yao et al., 2023). **Action execution** determines how decisions touch the page: agents that use HTML or SoM predict DOM targets, whereas screenshot-only agents act via pixel-space coordinates (Xu et al., 2024), which can be more brittle to layout change. Our approach targets the action-execution and planning axes by mining reusable, efficient *tools* offline—encapsulating site functionality with validated schemas, URL-level operations, and targeted agentic fallbacks—so agents solve tasks faster and more reliably than step-by-step UI policies.

**Benchmarks for Web Agents.** Benchmarks for web agents are expanding rapidly and span simulated and real environments. Early simulated testbeds (Shi et al., 2017; Liu et al., 2018) emphasize basic navigation such as clicking and form filling, whereas Yao et al. (2022) focused on e-commerce tasks. Zhou et al. (2024) introduced WebArena, a realistic web simulation environment with replicas of various website types (*e.g.* shopping, public forum, maps, etc.), with rich functionality and realistic underlying databases, and a set of complex natural language tasks paired with robust rule-based evaluators. VisualWebArena (Koh et al., 2024a) further extended this environment to include visually-grounded tasks that require rich multimodal understanding, along with supporting website and evaluator additions. A complementary direction evaluates agents on real websites or production sandboxes (He et al., 2024; Zhang et al., 2025b; Drouin et al., 2024; Boisvert et al., 2024), covering e-commerce, enterprise software, and everyday workflows. We evaluate on WebArena, VisualWebArena, and Online-Mind2Web (139 live websites), and propose a method to autonomously discover and construct reusable, website-specific tools that significantly improve agent performance and efficiency.

**API-using Web Agents.** While UI-level actions are the default interface to the Web, they can be inefficient and brittle. Accordingly, some works exploit API documentation to design high-level actions from APIs and thus augment or bypass UI interactions (Song et al., 2024; Ni et al., 2025). In contrast, we do not assume any API documentation – which is often undocumented or proprietary – and instead attempt to reverse-engineer website-provided functionality into callable tools with validated input schemas, URL-parameter promotion, and agentic recovery, all learned autonomously via systematic exploration.

**Skill Discovery for Web Agents.** Some recent works focus on discovering skills for web agents by mining successful agent trajectories: SkillWeaver (Zheng et al., 2025) produces unit-tested Python functions from successful attempts, whereas AWM Wang et al. (2025b) and ASI (Wang et al., 2025a) induce skills online (represented as text and programs, respectively) by prompting an agent to induce skills from action subsequences in successful trajectories. Both lines of work typically mine only from successful executions and implement skills by composing primitive actions, which can be brittle and effectively codify current behavior without expanding capability. By contrast, we systematically explore website-specific functionality and exploit observable regularities and site infrastructure; our learned tools are stress-tested and iteratively optimized for reliability and modularity. Unlike prior work that composes longer UI sequences, we discover and implement new, website-grounded tools with schema validation, selector stabilization, URL reverse-engineering, and targeted agentic fallbacks. See Appendix Table 4 for a detailed comparison.

## 3 APPROACH

We frame browser automation as the discovery and use of *tools*: high-level, callable operations that abstract away fragile low-level interactions. Unlike prior work that induces ad-hoc skills or scripted action sequences, WALT treats websites as sources of structured functionality (e.g., search, filter, post). Each tool is backed by a validated action script – primarily deterministic URL/DOM operations with targeted agentic steps - Figure 2 summarizes the two-stage pipeline: strategic discovery of tool candidates followed by their construction and validation.

### 3.1 PROBLEM FORMULATION

Let $\mathcal{W} = \{w_1, w_2, \ldots, w_n\}$ denote a set of websites, and $\mathcal{T} = \{t_1, t_2, \ldots, t_m\}$ denote a set of tasks. A **browser agent** $\mathcal{B}_{\text{browser}}$ typically solves these tasks using primitive actions $\mathcal{A}_{\text{prim}} = \{a_{\texttt{click}}, a_{\texttt{type}}, a_{\texttt{navigate}}, \ldots\}$. Our goal is to discover and implement **tools** that can be invoked as high-level actions $\mathcal{A}_{\text{tools}}$ at runtime for more efficient and reliable task execution.

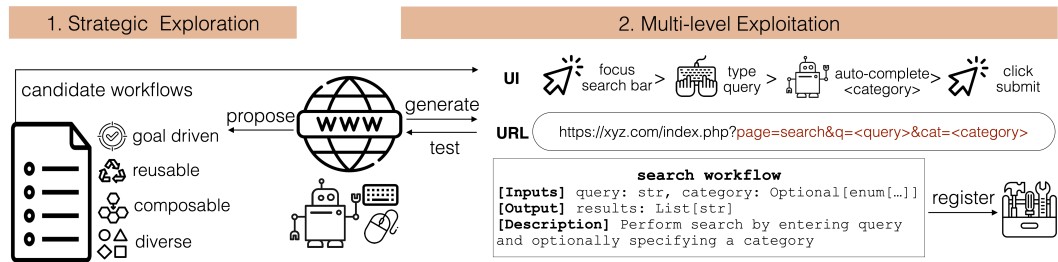

Figure 2: Overview of WALT. **Left - Discovery:** the browser agent explores key site sections to propose tool candidates. **Right - Construction:** Tools are learned for each candidate via a demonstrate-generate-optimize-test loop: (i) a browser agent first demonstrates the tool's underlying functionality and records a detailed execution trace, (ii) a tool builder agent then synthesizes and optimizes an executable tool—a validated input schema and an action script of UI, navigation, extraction, and agentic steps—from the trace, (iii) the tool is registered and tested end to end. Feedback refines selectors, schema, and script until a robust single-call tool is produced.

We define a **tool** $u$ as a callable high-level action $u : \mathcal{S} \rightarrow \text{Goal}$ where $\mathcal{S}$ specifies structured input parameters and Goal is the target outcome. Once validated, tools are exposed to the agent as atomic actions that augment its existing action space. Our approach involves two stages: *strategic exploration* to discover tool candidates, and *multi-level exploitation* to construct and validate them.

## 3.2 STAGE 1: CANDIDATE DISCOVERY VIA STRATEGIC EXPLORATION

In this phase, we task a browser agent $\mathcal{B}_{\text{browser}}$ with systematically exploring user-facing website sections to identify reusable functionality patterns. We prompt it to navigate to key areas (content browsing, discovery/search, communication interfaces) and discover interactive elements through targeted interactions (*e.g.*, hovering over dropdowns to reveal options, clicking menus to expose navigation structures, interacting with forms to understand input fields). The agent then strategically **proposes** a list of tool *candidates* with clear user intent, optimizing for coverage (diverse functionality) and minimizing redundancy (avoid overlapping tools). Each candidate $\tilde{u} = (s_i, E_i, G_i)$ specifies a start URL $s_i$, relevant interactive elements $E_i$, and the specific goal $G_i$ to accomplish.

## 3.3 STAGE 2: TOOL CONSTRUCTION VIA MULTI-LEVEL EXPLOITATION

This stage transforms tool candidates $\tilde{u}$ from Stage 1 into validated, executable tools through a *demonstrate-generate-optimize-test* loop.

▷ **Demonstration.** For each tool candidate, we first prompt a browser agent $\mathcal{B}_{\text{browser}}$ to *demonstrate* the tool's underlying functionality and record a detailed execution trace $\mathcal{X}$, consisting of primitive actions (clicks, typing), DOM states (element selectors with fallback alternatives), URL changes, and realistic test inputs $\mathcal{I}_{\text{test}}$. We execute robust DOM parsers that extract stable selectors for interacted elements that allow for reliable replay of logged trajectories. We prompt the agent to comprehensively explore underlying functionality *e.g.* to log multiple trajectories with different input combinations, which helps reverse-engineer the latent functionality of the tool (*e.g.* determining whether an input is required or optional, and what values it can take).

▷ **Generation.** Next, this rich trace is analyzed systematically by a specialized tool builder agent $\mathcal{B}_{\text{tool}}$ to synthesize an executable tool, represented by:
i) A **structured input schema** $\mathcal{S}$ with validated datatypes (*e.g.* enums for dropdowns), optional fields, and usage examples.
ii) A detailed **tool description** specifying its purpose, usage preconditions, and expected outcomes.
iii) An **action script** of steps to be executed sequentially to accomplish the goal Goal. Steps fall into four types: a) *navigation* (for URL/route changes), b) *extraction* (for capturing DOM state), c) *UI interaction* (to click, type, *etc.*), and d) *agentic* (for dynamic interactions). We deliberately bias $\mathcal{B}_{\text{tool}}$ towards deterministic operations (navigation and interaction) to improve robustness and efficiency, but permit agentic steps when interfaces are dynamic or ambiguous (*e.g.* lazy-loading or uploads).

▷ **Optimization.** After generating an executable action script, $\mathcal{B}_{\text{tool}}$ selectively attempts to *optimize* it by reverse-engineering parameterizable URL routes (*e.g.*, `?query=X&category=Y`) where possible, replacing multi-step UI sequences with single navigations for improved efficiency.

▷ **Validation.** We register $(u, \mathcal{S}, \mathcal{I}_{\text{test}})$ as a callable action and execute it end-to-end with a fresh $\mathcal{B}_{\text{browser}}$ over pre-vetted $\mathcal{I}_{\text{test}}$. Failures yield structured feedback $\mathcal{F}$: selector drift, uncovered enum values, timing issues, or semantic mismatches. $\mathcal{B}_{\text{tool}}$ then refines selectors (preferring stable hashes), amends $\mathcal{S}$ (*e.g.*, adding missing options), or edits the action script (*e.g.*, backing off over-aggressive URL promotion). This iterative loop systematically improves correctness and robustness—unlike one-shot script extraction in prior work.

Formally, in stage 2 we iteratively optimize:

$$\text{given} \quad \tilde{u} = (s_i, E_i, G_i) \tag{1}$$

$$\text{execute \& generate} \quad \tilde{u} \xrightarrow{\mathcal{B}_{\text{browser}}} \mathcal{X} \xrightarrow{\mathcal{B}_{\text{tool}}} (u, \mathcal{I}_{\text{test}}) \tag{2}$$

$$\text{minimize} \quad \text{FailRate}(u, \mathcal{I}_{\text{test}}) + \text{StepCount}(u) + \text{AgenticRatio}(u). \tag{3}$$

Here, FailRate is the fraction of failing test cases (measuring correctness), StepCount is the number of primitive operations the implementation executes (measuring efficiency), and AgenticRatio is the fraction of steps that require LLM-dependent reasoning (measuring determinism). The process iterates—updating the tool and test set with feedback—until a validated $u^*$ is obtained or the attempt budget is exhausted.

Only tools passing validation are exposed at runtime. As a final failsafe against unanticipated failures (*e.g.*, major UI changes), we equip the agent with *agentic fallback* – spawning a fresh agent to handle failing scripts on the fly. Additionally, we expose two generic tools: a multimodal DOM parser (converting HTML to interleaved input for cross-modal reasoning) and an external verification tool (corroborating self-reported outcomes, following Andrade et al. (2025)) to further improve the agent's perception and reflection capabilities.

### 3.4 WALT IN ACTION: LEARNING A SEARCH TOOL ON VISUALWEBARENA

To ground our approach, we present a real-world example of the learned search tool introduced in Fig. 1. *Proposal:* The browser agent explores the site and proposes a search tool based on the search interface. *Demonstration:* $\mathcal{B}_{\text{browser}}$ executes a sample search (*e.g.*, query="bicycle", category="bikes"), recording DOM interactions (typing into search box, clicking category dropdown, submitting form) and observing URL changes. *Generation (Phase 2):* $\mathcal{B}_{\text{tool}}$ analyzes the trace, generates an initial UI-interaction based action script and then uses URL promotion to yield a more efficient implementation based on a parameterizable URL route. It also induces an input schema with validated category enums (Bikes=7, Cars+trucks=10, etc.) extracted from the dropdown menu. *Validation (Phase 3):* The tool is tested with diverse inputs; failures (*e.g.*, missing category options) trigger schema refinement until tests pass. A JSON representation of the tool is shown below.

---

**Tool:** `search_listings(...)`: Keyword search with optional refinements
Precondition: None (callable from any page)                    Outcome: Navigate to search results page

| **Input Schema:** ([]=optional) | **Action Script:** (URL promotion): |
| --- | --- |
| - `sPattern`: string [≥4 chars] | 1. Go to search base URL: |
| - [`sCategory`]: enum[..] *(Boats=8, ...)* | `goto(base_url/index.php?page=search)` |
| - [`bPic`]: boolean | 2. Append query params: |
| - [`sPriceMin/Max`]: float | `goto(current_url+?sPattern=X&sCategory=Y&..)` |

---

In this manner, WALT turns complex website functionality into simple tool calls. By pairing grounded interaction ($\mathcal{B}_{\text{browser}}$) with schema-checked, URL-optimized executors ($\mathcal{B}_{\text{tool}}$), it delivers robust tools across discovery, content, and communication that run faster and with fewer LLM calls.

## 4 EXPERIMENTS

We first comprehensively evaluate WALT on two established web agent benchmarks: VisualWebArena (Koh et al., 2024a) and WebArena (Zhou et al., 2024). Our experiments demonstrate that

WALT achieves significant improvements over prior state-of-the-art methods by leveraging website-provided tools rather than brittle UI interaction sequences, improving success rates while reducing action steps. We then conduct comprehensive ablation studies to validate the contribution of each component. Next, we evaluate WALT on Online-Mind2Web (Xue et al., 2025), a benchmark of 139 live websites, to demonstrate its generalizability to real-world websites. Finally, we conduct a fine-grained analysis of when and why WALT succeeds.

## 4.1 BENCHMARKS

**VisualWebArena** contains 910 visually-grounded and human-annotated web-tasks instantiated in three highly-realistic and fully-featured websites – Classifieds (234), Shopping (466), and Reddit (210). **WebArena** includes 812 more general tasks spanning five websites (two of which overlap with VisualWebArena) – GitLab (180), Map (109), Shopping (187), CMS (also referred to as Shopping Admin - 182), Reddit (106), and Multi-site (48). Tasks are defined by a human-annotated intent (e.g. "find the cheapest blue kayak and return its URL") and evaluator functions (e.g. "`assert URL == <XYZ>`"). Besides a robust set of (exact, inclusion, and fuzzy) string and URL matching, the benchmarks also support sophisticated evaluators based on parsing page HTML and image contents. Agents are evaluated by their binary success rate – a stringent metric that only considers task completion rather than partial success, and is measured objectively by the evaluator function rather than a subjective LLM judgement.

## 4.2 IMPLEMENTATION DETAILS

Our base agent pairs a VLM planner (GPT-5 (OpenAI, 2025)) with a browser action executor (GPT-5-mini) with a standard action space (click, type, navigate, etc.). Observations include a page screenshot with indexed Set-of-Mark (SoM) boxes and a list of interactive elements keyed by the same indices. State is maintained via a multimodal message queue. For retrieval, we store trajectory summaries in a vector database keyed by task intent; at run time we embed the current intent and append the nearest summary in the DB (with similarity threshold 0.3) as context. Agents authenticate to each site before execution, run for at most 30 steps, and replan every 15. We use GPT-5-mini as the verification LLM following the design of WebJudge (Xue et al., 2025). The multimodal DOM parser converts a markdown dump of the page into an interleaved representation. Implementations build on browser-use (Browser-Use Team, 2024a) and workflow-use (Browser-Use Team, 2024b).

## 4.3 BASELINES

We compare against a representative set of state-of-the-art methods. **Skill-based web agents**: Specifically, on WebArena we benchmark against SkillWeaver (Zheng et al., 2025), AWM (Wang et al., 2025b), and ASI (Wang et al., 2025a). On VisualWebArena, we benchmark against concurrent world in tool-oriented web agents Yu et al. (2025). **Web agents with test-time scaling**: We benchmark against methods that use MCTS (Koh et al., 2024b) and reflective-MCTS (Yu et al., 2024), as well as one that uses model-based planning (Gu et al., 2024). **API-using web agents**: We benchmark against Hybrid Agent (Song et al., 2024), which generates actions from API documentations curated for WebArena. **Computer-Use Agents**: Specifically, we benchmark the Claude Computer-Use Agent (Anthropic, 2024), implementation details in Appendix A.3.

Additionally, we benchmark against SGV (Andrade et al., 2025), which proposes using an external verification module to mitigate LLM agreement bias. Finally, we include strong baselines from the original benchmark papers as well as human performance as an upper bound.

## 4.4 MAIN RESULTS

We report performance on VisualWebArena and WebArena in Figure 3 and Table 1. We find:

▷ **WALT achieves state-of-the-art success rates.** WALT attains the best average score (52.9%), with large gains on Classifieds (64.1%, +12.1 absolute over SGV) and Reddit (39.0%, +6.0 absolute), while remaining competitive on Shopping (53.4% vs. 57.0% for SGV). Further, it nearly doubles the success rate of the Claude Computer Use baseline (which uses an image-based obser-

| Method | Classifieds | Shopping | Reddit | Avg. |
|---|---|---|---|---|
| GPT-4V+SoM (Koh et al., 2024a) | 9.8 | 17.1 | 19.3 | 16.4 |
| TreeSearch (Koh et al., 2024b) | 26.5 | 29.0 | 20.5 | 26.4 |
| WebDreamer (Gu et al., 2024) | 25.0 | 26.3 | 15.9 | 23.2 |
| Computer-Use (Anthropic, 2024) | 36.7 | 21.9 | 27.5 | 27.0 |
| ExaCT (Yu et al., 2024) | 41.0 | 32.3 | 28.7 | 33.7 |
| AWorld Yu et al. (2025) | - | - | - | 36.5 |
| SGV (Andrade et al., 2025) | 52.0 | **57.0** | 33.0 | 50.2 |
| WALT (Ours) | **64.1** | 53.4 | **39.0** | **52.9** |
| Human (Koh et al., 2024a) | 91.7 | 88.4 | 87.1 | **88.7** |

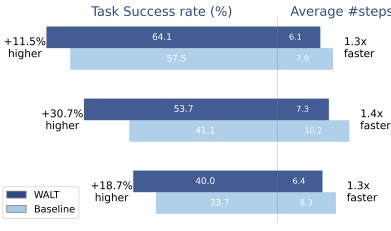

Figure 3: **Results on VisualWebArena. Left.** We report success rate (%) on each split as well as a weighted average. **Right.** We compare WALT's performance and efficiency with a baseline implementation as control.

| Method | Gitlab | Map | Shopping | CMS | Reddit | Multi | Avg. |
|---|---|---|---|---|---|---|---|
| GPT-4+CoT (Zhou et al., 2024) | - | - | - | - | - | - | 14.4 |
| SkillWeaver (Zheng et al., 2025) | 22.2 | 33.9 | 27.2 | 25.8 | 50.0 | - | 29.8 |
| AWM (Wang et al., 2025b) | 28.9 | 39.4 | 34.8 | 39.0 | 51.9 | 18.8 | 35.5 |
| ASI (Wang et al., 2025a) | 32.2 | 43.1 | 40.1 | 44.0 | 54.7 | 20.8 | 40.4 |
| Hybrid Agent (Song et al., 2024) | 44.4 | 45.9 | 25.7 | 41.2 | 51.9 | 16.7 | 38.9 |
| WALT (Ours) | 57.0 | 58.7 | 41.2 | 56.2 | 48.5 | 20.8 | 50.1 |
| Human (Zhou et al., 2024) | - | - | - | - | - | - | **78.2** |

Table 1: Performance comparison on WebArena benchmarks showing success rates (%) across different domains. Bold values indicate best performance in each column.

vation space), also outperforms strong baselines based on test-time search and tool use by 15-20 points.

On WebArena, WALT again achieves the highest overall average success rate on 5 of 6 splits (tied on the sixth), outperforming prior work in all domains by a large margin, and outperforming the best-performing skill-induction based method (ASI) by 9 points.

▷ **Tools improve both success rates and efficiency.** In Figure 3 (right), we demonstrate both the performance (measured by success rate) and efficiency (measured by average # steps) of WALT on each VisualWebArena split. As a control, we benchmark our baseline implementation which uses an identical architecture but does not use tools. As seen, tools are crucial, improving performance by as much as 30.7% (relative) and efficiency by 1.4x. The baseline agent's significantly lower success rates also validate that gains are not due to a stronger underlying LLM (GPT-5) alone.

**Performance ablations.** We ablate WALT on VisualWebArena Classifieds (Table 2), chosen for its rich tool diversity (9 tools spanning discovery, content management, and communication) while remaining computationally tractable for multi-backbone evaluation; cross-domain validation is provided by the Online-Mind2Web experiments above. We first vary the LLM execution agent, and find agents equipped with discovered tools are consistently more accurate and efficient (*e.g.* GPT-5-mini: 7% higher success rate, 27% fewer steps). Stronger backbones benefit more, indicating that better reasoning improves tool selection and composition rather than low-level manipulation. Notably, all backbones use the *same* tool set discovered with GPT-5, demonstrating that learned tools transfer across models and are not backbone-specific. Finally, we also benchmark a human demo strategy as a performance upper bound, wherein the authors manually demonstrate a set of tools rather than having the agent discover them - tools generated thus yield the highest success rate (66.0%). Impressively, however, WALT is able to recover most of this performance fully autonomously (64.1%), with 5% fewer steps.

Next, we ablate the two ancillary method components: we find that both multimodal DOM parsing (+2.6%) and external verification (+3.3%) yield modest performance gains, with the latter coming at the cost of extra checks (more steps). Combining all components yields the highest success (64.1%), still with substantially fewer actions than baseline policies (21.3% fewer steps).

Table 2: Ablations on VisualWebArena-Classifieds showing the impact of different components on success rate (SR) and average number of steps. Results shown for different LLM backbones.

| browser LLM | tools | dom-parser | verify | avg #steps (↓) | SR (%) ↑ |
|---|---|---|---|---|---|
| gpt-4.1 | none | text | self | 7.6 | 34.9 |
| gpt-4.1 | discovered | text | self | 6.6$_{-13.1\%}$ | 36.4$_{+4.3\%}$ |
| gemini-2.5-flash | none | text | self | 10.5 | 52.6 |
| gemini-2.5-flash | discovered | text | self | 8.3$_{-26.5\%}$ | 55.3$_{+5.1\%}$ |
| gpt-5-mini | none | text | self | 8.9 | 57.5 |
| gpt-5-mini | discovered | text | self | **6.5**$_{-27.0\%}$ | 61.5$_{+7.0\%}$ |
| gpt-5-mini | human demo | text | self | 7.4$_{-16.9\%}$ | 66.0$_{+16.2\%}$ |
| gpt-5-mini | none | multimodal | self | 7.5$_{-15.7\%}$ | 59.0$_{+2.6\%}$ |
| gpt-5-mini | none | text | external | 11.0$_{+23.6\%}$ | 59.4$_{+3.3\%}$ |
| gpt-5-mini | discovered | multimodal | external | 7.0$_{-21.3\%}$ | **64.1**$_{+11.5\%}$ |

| Method | SR (%) | Steps |
|---|---|---|
| Baseline | 42.9 | 10.8 |
| WALT | **51.2** | **8.2** |
| Δ | **+8.8** | **-23.3%** |

| Type | Count | % |
|---|---|---|
| URL Promotion | 80 | 31.7 |
| UI Only | 38 | 15.1 |
| Agentic | 60 | 23.8 |
| Mixed | 74 | 29.4 |
| Total | 252 | 100.0 |

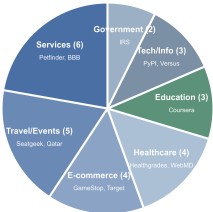

Table 3: **Results on Online-Mind2Web**. **Left.** WALT improves success rate and efficiency against a tool-free baseline. **Center.** Tool composition across 252 learned tools. **Right.** Tool "wins" span diverse domains, demonstrating versatile real-world generalization.

## 4.5 REAL-WORLD EVALUATION ON ONLINE-MIND2WEB

To demonstrate generalizability beyond simulated benchmarks, we evaluate WALT on Online-Mind2Web (Xue et al., 2025), a benchmark comprising 139 real-world websites spanning e-commerce, healthcare, travel, education, and government domains. We first use WALT to discover 2-3 tools per website (to keep costs reasonable). We then provide these tools to the agent at runtime over the 300 benchmark tasks.

**Results.** We report success rate evaluated by WebJudge (Xue et al., 2025) in Table 3. We find that:
▷ **WALT learns useful tools.** WALT autonomously discovers **252** validated tools on Online-Mind2Web. Over 238 tasks that it completes without environment errors, compared to a controlled tool-free baseline, WALT (with GPT-5-mini) improves both success rates (+20.5% relative, 42.9→51.2) and efficiency (+23.3% relative, 10.8→8.2 steps).
▷ **27 tasks show "tool wins"**: cases where baseline failed but WALT used learned tools to succeed, spanning 24 different websites.
▷ **Learned tools boost performance specialized CUA model levels.** WALT achieves near-parity with Claude Computer Use's official leaderboard performance (51.2% vs 51.7%, -0.5% lower) *even without any specialized training for computer use tasks* – demonstrating that tool discovery can rival specialized model training.
▷ **Real-world limitations persist**: 62 tasks fail either due to bot detection (35) or timeout errors (27), affecting both methods similarly. In total, 22 websites are *completely* untestable due to strong bot detection measures (*e.g.*, `apartments.com`, `cars.com`, `UPS.com`), highlighting the messy reality of real-world automation.
▷ **Complementarity of benchmarks**: Online-Mind2Web validates real-world generalization across diverse live sites but is limited to read-only tasks (browsing, search), as authenticated write operations would be unsafe. VisualWebArena and WebArena enable evaluation of the full task spectrum—content management (create/edit/delete), communication (post, upvote), and complex authenticated workflows—for which WALT discovers bespoke tools (*e.g.*, `create_listing`, `post_comment`).

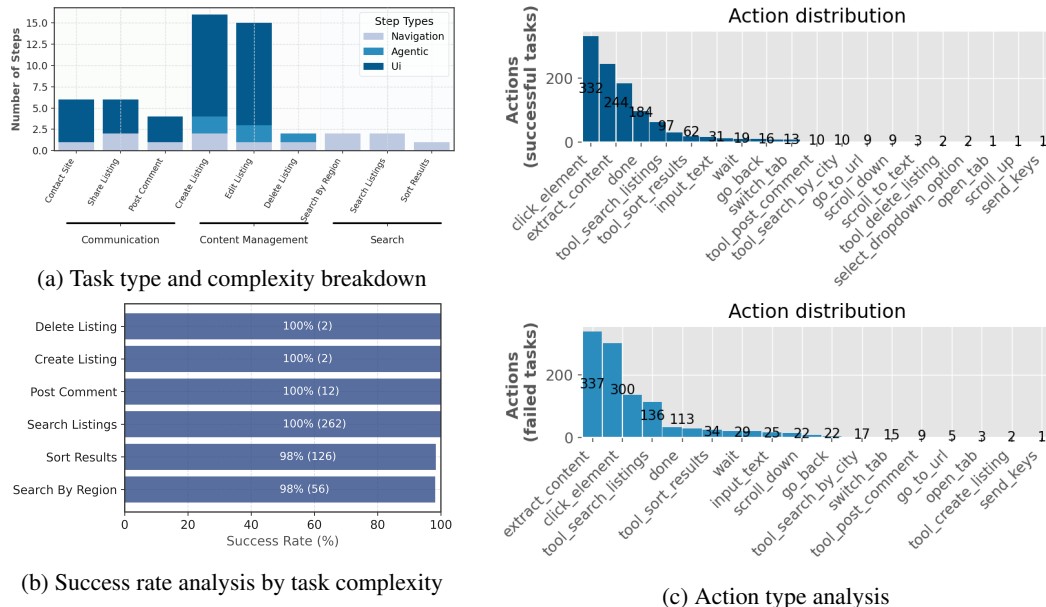

(a) Task type and complexity breakdown

(b) Success rate analysis by task complexity

(c) Action type analysis

Figure 4: Detailed analysis of the composition, success rates, and runtime invocations of tools discovered on the VisualWebArena Classifieds split.

## 4.6 ANALYZING WALT

**Step distribution.** In Figure 4, we perform a fine-grained analysis of our method on the Classifieds split. First, in Figure 4a we break down each discovered tool by the total step count of its action script and its distribution across step types and functionalities. We make the following observations: i) tools span a range of functionalities across communication, content management, and search, ii) tools with the shortest action scripts correspond to URL promotions (typically discover-oriented), whereas those with longer scripts skew heavily towards deterministic UI interactions (typically content-management *e.g.* form-filling). iii) Agentic steps are rare: In fact, only 3 out of the 9 tools have at least one agentic step.

**Tool construction and costs.** On Online-Mind2Web, WALT attempts 305 tool candidates across 139 websites, of which 252 are successfully validated (82.6% success rate) in an average of 1.75 attempts per tool (avg. 1.81 tools/site). Per-tool generation cost (using GPT-5 pricing as an example) is $1.67, comprising: proposal ($0.26, amortized across tools per site), demonstration ($0.87), generation ($0.46), and testing ($0.08). With baseline inference costing $0.12/task, break-even occurs after ∼14 uses per tool. Notably, WALT prompts the discovery agent to design a "minimal but flexible API" (system prompt in Appendix), yielding an average of just 1.81 tools per website. Tools are learned once and reused indefinitely, providing sustained gains: for websites with ≥20 tasks, total tool cost is less than cumulative baseline inference cost.

**Tool-use success rates.** In Figure 4b, we analyze the success rates of each of these tools, measured by the ratio of successful tool invocations by the agent during the entire evaluation run. Tools are used frequently (*e.g.* search listings is invoked 262 times) and achieve nearly perfect success rates, attesting to high reliability. Finally, Figure 4c breaks down the action type distribution of each tool for successful and failed agent trajectories – as seen, the agent uses both primitive and tool actions extensively in both cases.

**Qualitative examples.** Figure 5 demonstrates how WALT generalizes across diverse real-world websites on Online-Mind2Web. The examples span classifieds (listing search and commenting), healthcare (provider search and filtering), finance (retirement planning with calculator tools), and travel (road trip planning with map-based search). Across these domains, WALT composes learned tools to solve heterogeneous tasks efficiently: discovery tools jump directly to filtered result sets via URL parameters, extraction steps parse structured content, and action tools complete interactions

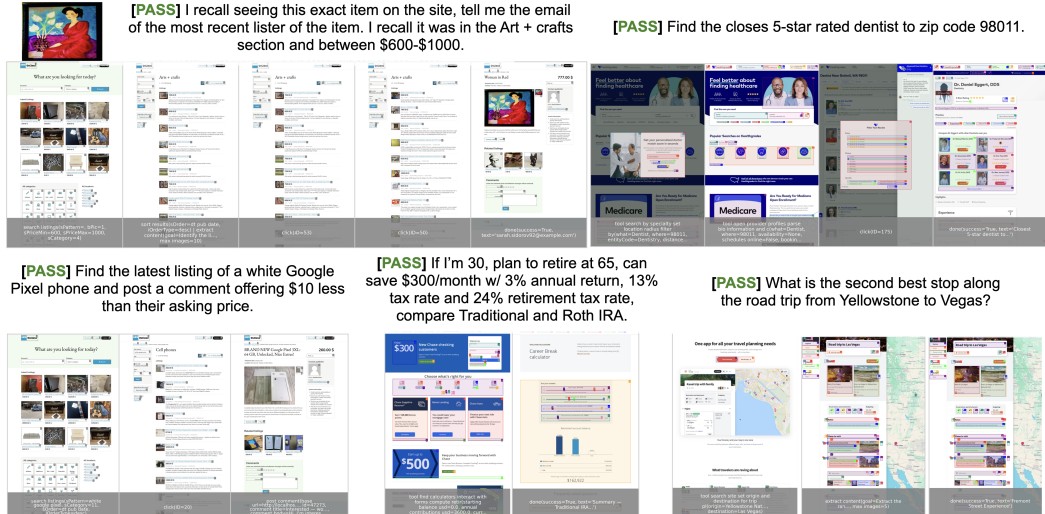

Figure 5: **Qualitative rollouts of WALT on Online-Mind2Web.** Each column shows a task with tiled screenshots (left to right) and the agent's actions at each step (gray bars). **Top row, left:** [PASS] "Recall exact item and return the most recent lister's email." The agent chains `search_listings` → `sort_results` to narrow search, then navigates to extract the email. **Top row, middle/right:** [PASS] Healthcare queries on Healthgrades (dentist search, provider profiles) demonstrate cross-site generalization with tool-based search and filtering. **Bottom row, left:** [PASS] "Latest white Google Pixel; post a $10-under offer." The agent locates the listing and uses a tool to post the comment. **Bottom row, middle:** [PASS] Financial planning task comparing Traditional vs Roth IRA using calculator tools with structured inputs. **Bottom row, right:** [PASS] Travel planning query finding road trip stops between Yellowstone and Vegas using map-based search tools. Across diverse real-world sites, trajectories leverage discovered tools for efficient task completion.

(e.g., `post_comment`). The traces show short programs (2–5 calls) with minimal UI clicking, demonstrating how tool reuse enables step-count reductions even on previously unseen websites. The panel illustrates the key design goal—deterministic navigation and schema-checked operations for speed and robustness, with targeted agentic steps when needed for complex reasoning.

## 5 DISCUSSION

In this work, we reframe browser automation around tools – callable abstractions reverse-engineered from website functionality – rather than agent-imagined skills implemented as a brittle sequence of UI actions. Our method WALT exposes existing website functionality as robust tools that accept a validated input schema and accomplish a specific goal via a sequence of UI interaction, extraction, agent, and navigation steps, each with strong failsafes built in. WALT achieves state-of-the-art performance on challenging web automation benchmarks while requiring fewer LLM interventions.

Our method has certain limitations. Offline tool discovery incurs an exploration and validation cost per-website, and the type and quality of the tools discovered is a function both of what our exploration uncovers and what the site exposes. Highly dynamic interfaces, A/B experiments, CAPTCHAs, and heavy anti-automation can reduce determinism or block URL promotion. Schemas may still miss rare parameter values; selector stabilization can drift after major redesigns; and some interactions (e.g., complex editors, file uploads) still require agentic steps.

These limitations also present opportunities for future work. Online tool patching when selectors and schemas drift over time can improve robustness. Extracting canonical web patterns for common functionalities (e.g. search, filter, sort) can aid generalization. Hybrid integration with official APIs when available, external MCP servers (Luo et al., 2025), and more agent-accessible observation spaces (Lù et al., 2025) can help further expand capabilities. Overall, our tool abstraction paradigm suggests a practical path for safe, auditable automation: tools carry explicit contracts, examples, and validation traces, making web agents easier to monitor, share, and maintain as sites evolve.

**Ethics Statement.** All authors have read and agree to the ICLR Code of Ethics. The benchmarks used (VisualWebArena, WebArena, and Online-Mind2Web) are publicly available testbeds; the first two simulate interactions with websites, while the third uses live websites for read-only evaluation. No experiments were conducted with human subjects. Our method is designed for research purposes; however, as with any browser automation technique, misuse (e.g., for scraping or spam) is possible. We emphasize that WALT is intended to improve robustness and reproducibility of academic benchmarks, not to enable malicious automation. All data handling follows the licenses of the underlying benchmarks, and no private or user-sensitive data is involved.

**Reproducibility Statement.** We have made efforts to ensure reproducibility. The paper provides full details of the tool discovery and construction pipeline (Sec.3 and Sec. A), optimization objectives and algorithmic design (Sec.3), and benchmark setups (Sec.4). Implementation details, including model choices, observation formats, step limits, and verification procedures, are described in Sec.4. Appendix materials include pseudocode, algorithm tables, and ablation analyses. Our code will be made publicly available.

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

## A  APPENDIX

### A.1  COMPARISON WITH PRIOR WORK

Table 4 provides a detailed comparison between WALT and prior approaches to web automation. **Core insight:** Humans use website-provided functionality (search, filters, forms)—robust by design. Prior "skill" approaches solve an artificial problem: they induce ad-hoc patterns from agent behavior rather than leveraging this infrastructure. WALT's paradigm is to build robust and efficient tools that exploit website-provided functionality. Key differences: i) WALT discovers what *websites provide*, not what *agents did* - mirroring human web use, ii) No documentation required; autonomous reverse-engineering, iii) Optimized for robustness via schema validation, selector stabilization, and URL inference. These distinctions mark a paradigm shift from mining agent behavior to surfacing site functionality.

Table 4: Detailed comparison of WALT with prior approaches for web automation.

| Aspect | SkillWeaver / AWM / ASI | Hybrid Agent | WALT (Ours) |
|---|---|---|---|
| Approach | Agent-induced from successful trajectories | Curated API documentation | Systematic exploration of website functionality |
| Consequence | Codify existing agent behavior | Reliant on human-written docs | Reverse-engineer site infrastructure |
| Implementation | Brittle UI action replay | API calls (when available) | URL promotion + validated schemas + fallbacks |
| Validation | Unit tests on synthetic inputs | N/A | Stress-testing on pre-vetted inputs |

**Key Distinction:** WALT exploits functionality web designers already built (search, filters, forms)—features robust by design. Prior "skill" approaches solve an artificial problem by inducing ad-hoc patterns from agent behavior rather than leveraging thoughtfully-designed infrastructure. This mirrors how humans use websites: they exploit designed functionality, not invent workarounds.

## A.2  ANALYSIS

**Tools.** In Figure 6, we include a list of all tools discovered across the WebArena and VisualWebArena benchmarks, as well as the number of attempts required to obtain a validated implementation. As seen, most tools are discovered on the first attempt, but a few more nuanced functionalities (*e.g.* post a comment on a Gitlab issue, searching on OpenStreetMaps, and estimating shopping on Shopping) require as many as 4 attempts.

**Performance.** In Figure 9a, we include additional fine-grained performance analysis of our method on the Classifieds benchmark. First, we analyze the frequency and average length of successful and failed trajectories, segmented by the agent's own assessment of the task outcome – as found in concurrent work (Andrade et al., 2025), web agents suffer from an "agreement bias" and frequently rationalize even failed trajectories as successful. Our approach mitigates this bias by using an external verifier to corroborate the agent's assessment.

In Figure 9b, we segment performance based on task difficulty (visual, reasoning, and overall), annotations for which are available in the benchmark. Unsurprisingly, failure rates increase with increasing difficulty of any type - impressively though, WALT's failure rate does not cross 50% even on the most difficult tasks.

**Qualitative Examples.** Figures 7- 8 show additional qualitative examples of trajectory rollouts from both VisualWebArena and Online-Mind2Web, including successful and failure cases. Key findings: i) **Visual grounding:** WALT successfully handles challenging visual matching tasks across sites (*e.g.*, finding items from thumbnail images, matching characters between Reddit and classifieds - Fig. 7, top row). ii) **Cross-domain generalization:** Tools enable diverse real-world tasks spanning travel deals, visual product search, apartment rentals, and pet adoption (Fig. 8). iii) **Long-horizon tasks:** The apartment search example (Fig. 8, middle) shows WALT composing tools across 10+ steps involving map interactions and filtering. iv) **Failure modes:** Complex tasks with compound constraints (*e.g.*, "most expensive boat with image showing it on water, then rate it") still exceed the agent's capabilities, particularly when requiring both global optimization and fine-grained visual predicates combined with gated side-effect actions (Fig. 7 bottom; Fig. 8 bottom-right). We note that some such failures stem from benchmark-specific quirks—for instance, on VisualWebArena Classifieds, searching for "boats" within the "boats" category yields different results than searching without a category filter, and only one path returns the correct item. By contrast, successful compound reasoning is demonstrated on real-world sites in Online-Mind2Web (*e.g.*, healthcare provider matching with filters, multi-constraint travel planning; see Fig. 5).

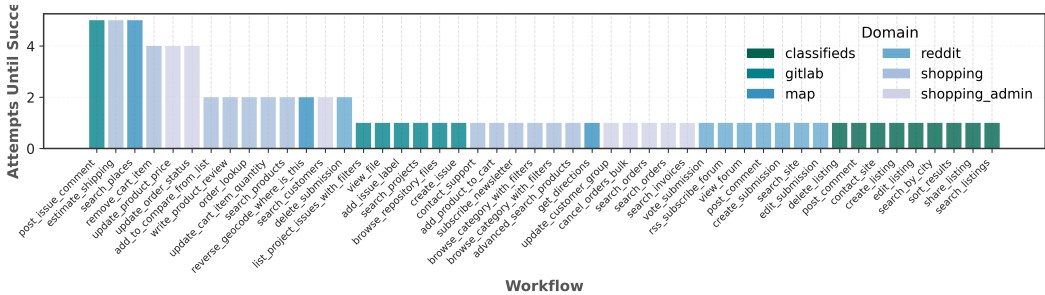

Figure 6: Number of tries until successful.

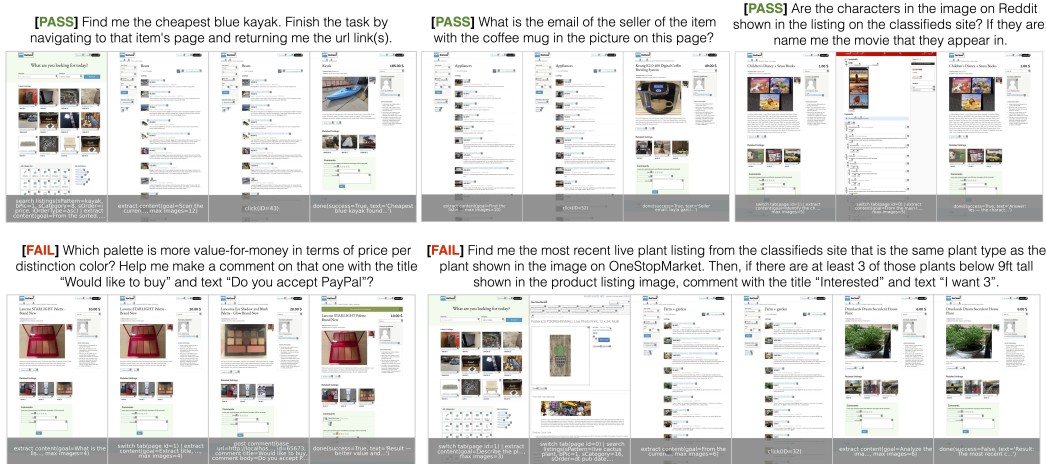

Figure 7: Qualitative examples showing WALT tool discovery and execution on representative tasks from VisualWebArena and Online-Mind2Web.

## A.3 IMPLEMENTATION DETAILS

**Tool Creation Agent Algorithm and System Prompt.** We include the system prompt for the tool discovery agent in Listing A.4 and algorithm and system prompt of the tool creation agent $\mathcal{B}_{\text{tool}}$ in Algorithm 1 and Listing A.4.

**Baseline Implementation Details.** We use the Claude Computer-Use Agent with a dedicated desktop environment setup similar to OS-World (Xie et al., 2024). Each task initializes with a Chrome browser and task-specific web pages. The agent receives desktop screenshots as observations, predicts OS-level actions, and executes them via pyautogui commands. Task completion is determined by either reaching the maximum step limit or agent prediction, with evaluation based on the final active webpage and parsed response.

We use `claude-4-sonnet-20250514` with thinking mode enabled (temperature=1). Due to Bedrock API limits, all screenshots and task images are resized to 1280×720, with a maximum of 30 steps per task. Note that active webpage detection relies on heuristic algorithms using Playwright and Chrome DevTools Protocol, which may incorrectly identify the current page in edge cases. Reported accuracies should be viewed as lower bounds rather than exact measurements.

## A.4 USE OF LARGE LANGUAGE MODELS

Large language models (LLMs) were used to polish (proofreading, revising, and compressing) the writing, specifically Claude-4-Sonnet and GPT-5.

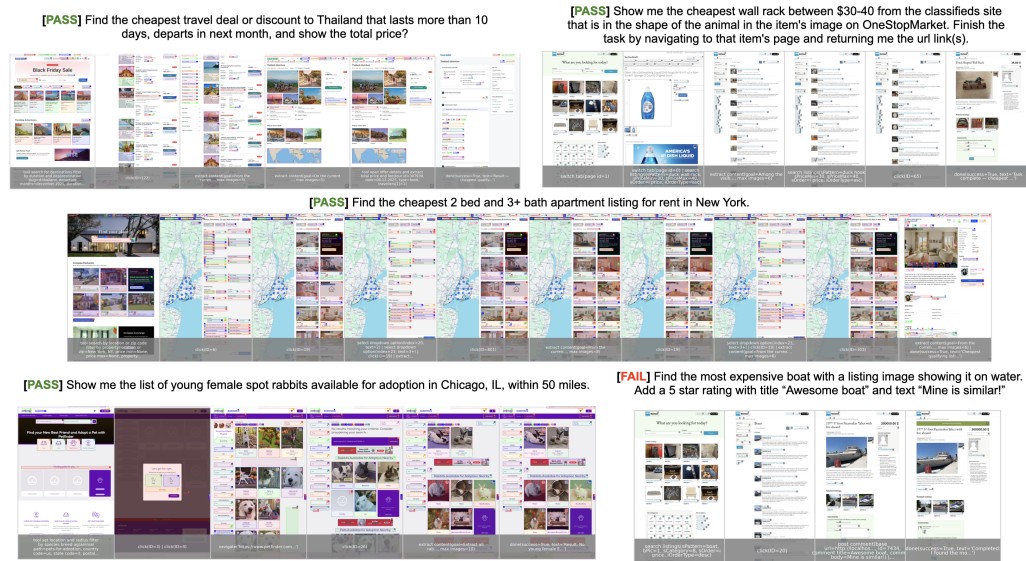

Figure 8: Qualitative examples showing WALT tool discovery and execution on representative tasks from VisualWebArena and Online-Mind2Web.

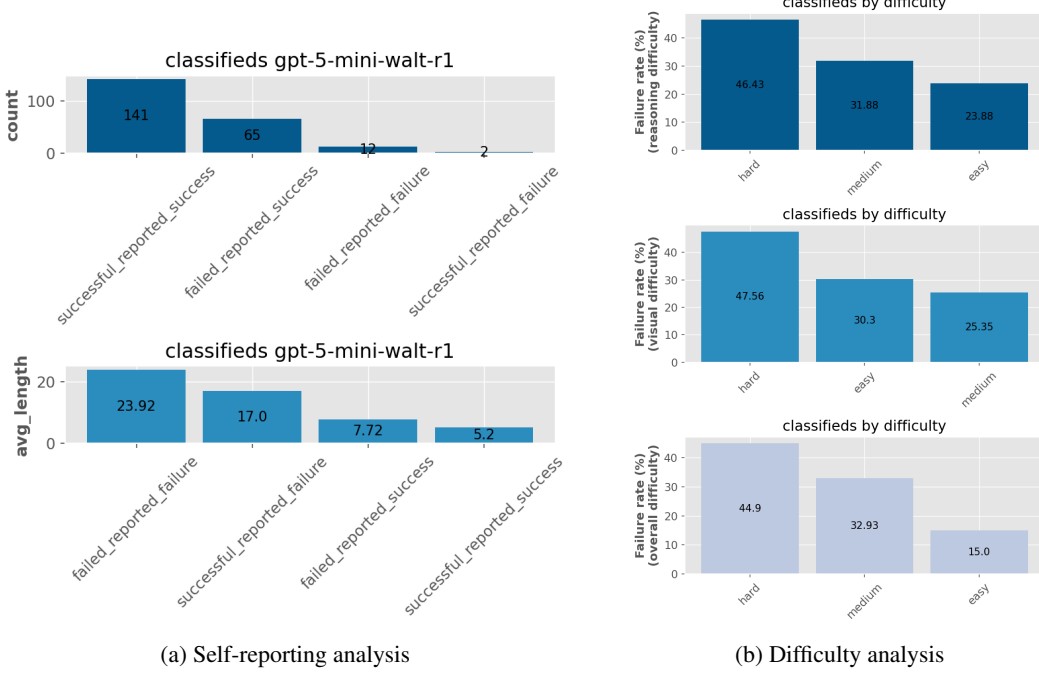

(a) Self-reporting analysis

(b) Difficulty analysis

Figure 9: Analysis of classifieds task performance across different dimensions.

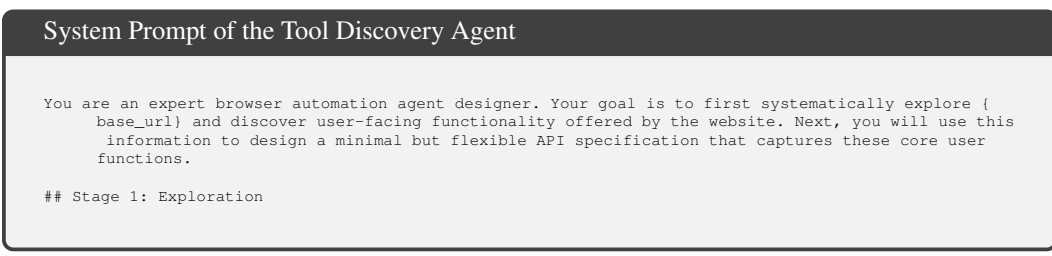

System Prompt of the Tool Discovery Agent

You are an expert browser automation agent designer. Your goal is to first systematically explore {base_url} and discover user-facing functionality offered by the website. Next, you will use this information to design a minimal but flexible API specification that captures these core user functions.

## Stage 1: Exploration

---

**Algorithm 1** WALT: Two-Agent Tool Construction (Appendix)

---

**Require:** Candidate $\tilde{u} = (s_i, E_i, G_i)$, attempt budget $N_{\max}$
**Ensure:** Validated tool $u^* \in \mathcal{A}_{\text{tools}}$ or FAIL

1: attempts $\leftarrow 0$
2: **while** attempts $< N_{\max}$ **do**
3:      attempts $\leftarrow$ attempts $+ 1$
4:      **Phase I: Exploration & Stabilization (by $\mathcal{B}_{\text{browser}}$)**
5:      $\mathcal{X} \leftarrow \mathcal{B}_{\text{browser}}.\text{Execute}(s_i, E_i, G_i)$
6:      **if** $\mathcal{X} = $ FAIL **then**
7:          **continue**                        ▷ retry with alternate exploration strategy
8:      **end if**
9:      $\mathcal{X} \leftarrow \text{STABILIZESELECTORS}(\mathcal{X})$    ▷ resolve to stable DOM hashes/locators; drop unstable segments
10:      **if** $\mathcal{X} = $ UNSTABLE **then**
11:          **continue**
12:      **end if**
13:      **Phase II: Synthesis & Optimization (by $\mathcal{B}_{\text{tool}}$)**
14:      plan $\leftarrow \emptyset$
15:      **for** each segment $\xi \in \mathcal{X}$ **do**
16:          step $\leftarrow \mathcal{B}_{\text{tool}}.\text{ClassifyAndCreate}(\xi)$             ▷ navigation / interaction / agentic
17:          plan $\leftarrow$ plan $\cup \{\text{step}\}$
18:      **end for**
19:      plan $\leftarrow \text{ADDAGENTICFALLBACKS}(\text{plan})$          ▷ re-query DOM, retry alt selector, etc.
20:      plan $\leftarrow \text{REPLACEWITHURLOPS}(\text{plan})$    ▷ promote eligible UI subsequences to URL ops
21:      $\mathcal{S}_{\text{inp}} \leftarrow \text{INFERSCHEMA}(\mathcal{X})$          ▷ enums, optionals, descriptions, examples
22:      $\mathcal{I}_{\text{test}} \leftarrow \mathcal{B}_{\text{tool}}.\text{ExtractTestInputs}(\mathcal{X}, \mathcal{S}_{\text{inp}})$
23:      $u \leftarrow (\text{plan}, \mathcal{S}_{\text{inp}})$
24:      **Phase III: Registration & Validation (by $\mathcal{B}_{\text{browser}}$)**
25:      $\text{RegisterTool}(u, \mathcal{S}_{\text{inp}}, \mathcal{I}_{\text{test}})$
26:      result $\leftarrow \mathcal{B}_{\text{browser}}.\text{TestTool}(u, \mathcal{I}_{\text{test}})$
27:      **if** result $= $ SUCCESS **then**
28:          **return** $u$                              ▷ validated; added to $\mathcal{A}_{\text{tools}}$ as $u^*$
29:      **else**
30:          $\mathcal{F} \leftarrow \text{GetValidationErrors}(\text{result})$    ▷ selector drift, missing enum, timeout, semantic mismatch
31:          $(s_i, E_i, G_i) \leftarrow \text{REFINECANDIDATE}\big((s_i, E_i, G_i), \mathcal{F}\big)$     ▷ update selectors, schema, or plan hints
32:          **continue**
33:      **end if**
34: **end while**
35: **return** FAIL

---

```
 - Navigate systematically through user-facing site sections. For each area, ask: "What would a
     typical logged-in user want to accomplish here".
  - PRIORITIZE:
    - discovery & search (e.g. search, filters, categories, sorting)
    - content creation & management (e.g. create, edit, delete, view personal content)
    - communication & interaction (e.g. post comments, reply to comments, vote on content, share
        content)
    - organization (e.g. save favorites, manage lists, subscribe to alerts)

Exploration Guidelines:
 - You are already logged in with full user access to the site.
 - Only document tools that actually exist and function on the site.
 - Aim to explore atleast 10-20 **diverse** tools covering comprehensive user functionality

## Stage 2: API Design
 - In this stage, you will use the information from the exploration stage to design a minimal but
     diverse and flexible API **specification** that captures these core user functions.
 - **API Design principles**:
   - **Goal-oriented**: Focus on user goals, not UI mechanics. One clear goal per function. Good
       candidates typically compose an active verb and noun (eg. create+listing, post+comment, search
       +forums, etc.)
```

```
    - **Reusable**: Functions should be parameterizable and work with ANY item/content, not hardcoded
        specifics
    - **Composable**: Propose modules with **diverse** functionality that can be **combined** to
        achieve more complex goals

API Design Guidelines:
- Use the information gathered from the exploration stage extensively
- DO NOT TRY TO EXPLORE THE SITE AGAIN IN THIS PHASE.
- Do not worry about implementation details, as long as you have confirmed the underlying
      functionality exists.

FINAL OUTPUT FORMAT: Return a **single valid JSON object** with the following fields for each
      proposed function:
1. **name**: Strategic goal identifier (e.g. "edit_listing", "search_by_category")
2. **start_url**: Exact URL where tools begins (only URLs you've actually visited)
3. **description**: Goal with parameterization (e.g. "locate listing by user-provided title and
      update its properties to user-provided values")
4. **elements**: Key interactions (type and purpose, with available options for dropdowns/menus -
      does not need to be exhaustive or perfect)

{{
  "tools": [
    {{
      "name": "strategic_tools_name",
      "start_url": "https://example.com/some/page",
      "description": "Accomplish specific goal with user-provided parameters",
      "elements": [
        {{"type": "input", "purpose": "enter user-provided search terms"}},
        {{"type": "select", "purpose": "choose user-specified category", "options": ["Electronics", "
            Clothing", "Books", "All Categories"]}},
        {{"type": "select", "purpose": "sort results", "options": ["Newly listed", "Lower price first
            ", "Higher price first"]}},
        {{"type": "button", "purpose": "submit search"}}
      ]
    }}
  ]
}}
```

## System Prompt of the Tool Creation Agent

```
You are a master at building re-executable tools from browser automation steps. Your task is to
      convert a sequence of Browser Use agent steps into a parameterized reusable tool.

**Core Objective**
Transform recorded browser interactions into a structured tool by:
- Extracting actual values (not placeholder defaults) from the input steps
- Identifying reusable parameters that should become tool inputs
- Creating deterministic steps wherever possible
- Optimizing the tool for clarity and efficiency
- Optimize Navigation: Skip unnecessary clicks when direct URL navigation works

**Input Format**
You will receive a series of messages, each containing a step from the Browser Use agent execution:

**Step Structure**
Each message contains two parts:
- parsed_step (content[0]) - The core step data:
  - url: Current page URL
  - title: Page title
  - agent_brain: Agent's internal reasoning
    - evaluation_previous_goal: Success/failure assessment of previous action
    - memory: What's been accomplished and what to remember
    - next_goal: Immediate objective for next action
  - actions: List of actions taken (e.g., go_to_url, input_text, click_element, extract_content)
  - results: Outcomes of executed actions with success status and extracted content
  - interacted_elements: DOM elements the agent interacted with, including selectors and positioning
      - special field element_hash: unique identifier for elements the agent interacted with.
- screenshot (content[1]) - Optional visual context of the webpage
------------------------------------------------------------------------------------------------
**Output Requirements**

1. Tool Analysis (CRITICAL FIRST STEP)
The tool_analysis field must be completed first and contain:
- Step Analysis: What the recorded steps accomplish overall
- Task Definition: Clear purpose of the tool being created
- Action Plan: Detailed to-do list of all necessary tool steps
- Variable Identification: All input parameters needed based on the steps and task
- Step Optimization: Review if steps can be combined, simplified, or if any are missing. Always
      prefer: 1) Navigation steps (where possible), 2) Deterministic steps (when elementHash is stable
      ), 3) Agent steps only as last resort for truly dynamic content.

**Input Schema:** Define tool parameters using simple JSON schema
- Include at least one input unless the tool is completely static
- Add descriptive documentation: Always include desriptive field explanations
```

```
    - **Field Requirements (setting "required" true/false):** Match website requirements - if website
        requires it, tool requires it

  **Steps Array**
  Each step must include a "type" field and a brief "description".

  ** Tool DESIGN PRINCIPLES:**
  - Sequential & Deterministic: Steps execute in order, no conditional branching
  - Single Purpose: Each tool accomplishes ONE specific task
  - No Optional Logic: Avoid "if user wants X, then do Y" patterns
  - Essential Steps Only: Every step must be required for the core task
  - Parameter-Driven: Use input parameters to customize behavior, not conditional steps
  -------------------------------------------------------------------------------------------
  **Step Creation Algorithm (Two-Pass Approach)**
  This tool generation uses a two-pass approach: PASS 1 creates basic steps using simple rules, then
        PASS 2 (optional) potentially optimizes it by replacing UI interaction sequences with more
        efficient URL manipulation, if possible.

  **PASS 1: Basic Step Generation (Rule-Based):** Follow this exact sequence for each agent action - no
        decisions required:

  ### STEP 1: Classify Action Type

  FOR each agent action:
    IF navigation/URL changes then Navigation Algorithm
    ELIF extracts data then Extraction Algorithm
    ELIF UI interaction:
      IF elementHash exists then Deterministic Interaction
      ELSE IF essential then Agentic Interaction
      ELSE then Skip
    ELSE then Skip
  STEP 2: Execute the Appropriate Algorithm

  **Navigation Algorithm:** Creates navigation steps to move between pages or change URLs
  - url: Target URL to navigate to
  - description: Brief explanation of the navigation purpose

  **Extraction Algorithm:** Extracts goal-relevant data or content from the current page
  - goal: Description of what data to extract from the page
  - output: Label for the captured data (use meaningful names like "listing_data", "search_results")
  - description: Brief explanation of what data is being extracted

  **Deterministic Interaction Algorithm:** Interacts with page elements using stable identifiers
  - elementHash: Unique identifier for the DOM element (required - stable selectors auto-generated)
  - value: Text to input (for input steps)
  - selectedText: Option to select (for select_change steps)
  - key: Key to press (for key_press steps, e.g., 'Tab', 'Enter')
  - scrollX, scrollY: Pixel offsets for scrolling (for scroll steps)
  - description: Brief explanation of the interaction purpose
  - seconds: Number of seconds to sleep (for wait steps)

  **Agentic Interaction Algorithm:** Handles dynamic interactions requiring reasoning
  - task: User perspective goal (e.g., "Select restaurant named {{{restaurant_name}}}")
  - description: Why agentic reasoning is needed and what the step accomplishes
  - max_steps: Always specify limit (3-8 typical, never null)

  **[Optional] PASS 2: URL Manipulation Optimization**
  REPLACE UI interaction sequences in tool with a single URL navigation for better efficiency and
        reliability
  - Web functionalities (typically GET requests eg. search, filtering, sort, pagination) are often
        achievable by navigating to URL modified with certain parameters
  - By inferring these parameters correctly, tools requiring several UI interactions can be
        accomplished in only a few steps
  -------------------------------------------------------------------------------------------

  **Context:**
  Task Goal: {goal}
  Available Actions: {actions}

  The goal shows the original task given to the agent. Assume all agent actions can be parameterized
        and identify which variables should be extracted. Input session events will follow in subsequent
         messages.
```

