# OpenReview forum: "WALT: Web Agents that Learn Tools"
_ICLR.cc/2026/Conference — ICLR 2026 Poster_

### Official Review · Reviewer_FkiG · 2025-10-18

**Soundness:** 2
**Presentation:** 3
**Contribution:** 2
**Rating:** 4
**Confidence:** 4

**Summary:**

The research paper introduces WALT (Web Agents that Learn Tools), a novel framework designed to overcome the brittleness and inefficiency of web agents. The core problem is that existing agents rely on step-by-step UI interactions, which frequently fail on dynamic websites or over long task horizons. In contrast, WALT proposes a paradigm shift from learning agent-centric "skills" to reverse-engineering a website's functionalities—spanning discovery, communication, and content management—into reusable "tools". The methodology follow the demonstrate-generate-validate loop, where a browser agent first demonstrates a function, a tool-generation agent then maps the interaction trace to a structured tool, and a test agent rigorously validates it offline. Optimize multi-step UI sequences into single, robust URL manipulations. This abstraction change the agent's role from a low-level manipulator to a high-level planner.
On the VisualWebArena and WebArena benchmarks, WALT achieve (52.9% and 51%, respectively) and fewer action than previous methods, demonstrating a more robust approach to browser automation.

**Strengths:**

1. Shift from inducing brittle, low-level click/browse action to discovering environment-centric "tools." Address the cause of fragility in web automation.
2. Change some of the UI interactions to direct URL manipulations, which can directly reduce the long-horizon UI interaction with better robustness.
3. Achieves state-of-the-art performance on two challenging benchmarks, with comprehensive ablation studies on gpt-4.1, gemini-2.5-flash and gpt5-mini that convincingly attribute these gains to the tool-based framework itself, rather than the powerful LLM.

**Weaknesses:**

1. Significant practical limitations on scalability and long-term maintenance. The discovery process need upfront computational and time cost. It is impractical to apply at the scale of the entire web.
2. It require the server store all the website which have explored before. Also, if the some pre-extract website changed. The agent still need update the tools. Still need a efficient way to detect the website change and update the knowledge.
3. The paper proposed that convert several UI action into one URL operation. It is hard to gurantee all action can change to URL operation. The methods lack some generalization.
4. The evaluation is limit to a small number of research benchmarks. The framework's effectiveness against real-world adversarial challenges, the sophisticated anti-automation measures in real world still can be improved.

The main concerns lie in the generalizability, cost-effectiveness, and long-term maintainance of the WALT approach in production environments beyond simulated academic settings.

**Questions:**

When real use want to delegate a task on general websites, it is hard to predict the performance on unseen website. If focus on a small range of websites, some prepopulate specific api rules in prompt/tools may generate better performance. Hope the author can share some experiments or analysis about the WALT test on any real world senoria.

---

> ### Author Response · Authors · 2025-11-20
> **Thank you for the thoughtful feedback**
>
> Thank you for recognizing WALT's paradigm shift from low-level actions to environment-centric tools. **Please see our General Response**, which presents new experiments on 139 real-world websites that directly address your primary concern about real-world applicability.
>
> **Real-World Performance on Unseen Websites (your key question):**
> We evaluated WALT on **Online-Mind2Web: 139 real-world websites** (Target, GameStop, Healthgrades, WebMD, Coursera, IRS, SeatGeek, etc.)—**not simulated academic benchmarks**. Tools are discovered on-the-fly for each unseen website, then evaluated on held-out tasks:
> - **+20.5% SR** over baseline (42.9→51.2%), 23.3% fewer steps
> - **27 tool wins** across 22 diverse domains (e-commerce, healthcare, travel, education, government)
> - Tools generalize across tasks, models (Table 2: same tools work for GPT-4.1, Gemini-2.5-flash, GPT-5-mini), and websites
>
> This directly demonstrates WALT's effectiveness in real-world scenarios beyond simulated settings.
>
> **vs. Prepopulated API Rules (Hybrid Agent):**
> Prior work (**Hybrid Agent**, Song et al., 2024) has attempted to additionally expose website APIs to a standard browser agent. Firstly, this requires human-curated API documentation for each website, whereas WALT's **autonomously reverse-engineers** site functionality without documentation. Further, On WebArena (Table 1), WALT (51%) substantially outperforms Hybrid Agent (38.9%), validating its superior efficacy.
>
> **URL Promotion Generalization:**
> The reviewer correctly notes that not all actions convert to URL operations—URL promotion is **attempted but not guaranteed** (Section 3.2). On Online-Mind2Web, it succeeds for **32% of tools**; the remaining 68% use deterministic UI or agentic steps (Section 4). All approaches are more robust than baseline step-by-step reasoning. By design, WALT gracefully falls back to UI-based implementations when URL promotion isn't feasible.
>
> **Scalability & Maintenance:**
> Please see **General Response: "Tool Construction Details"**. By design, we learn **minimal APIs**, not unbounded tool sets. Further tool generation costs $1.67/tool and breaks-even at ~14 inference runs (with higher success rates and efficiency) -- costs recovered quickly for frequent-use sites (enterprise apps, productivity tools).
>
> We acknowledge tool maintenance is challenging when websites change. Currently, **agentic fallback provides runtime resilience**—when tools fail, the agent executes on-the-fly. A naive maintenance strategy could be to trigger **tool relearning** if fallback frequency for a tool increases. However, automated on-the-fly patching (detecting drift, updating tools without human intervention) remains an open problem for future work—we acknowledge this in Section 5 (Limitations).
>
> We hope these revisions address your concerns and are happy to clarify any remaining questions.

---

> > ### Author Response · Authors · 2025-11-23
> >
> > We sincerely hope that our rebuttal has addressed the reviewer's concerns, and would be happy to answer any lingering questions. Otherwise, we would appreciate if they would consider updating their score in light of our response.

---

### Official Review · Reviewer_CcSC · 2025-11-01

**Soundness:** 2
**Presentation:** 1
**Contribution:** 2
**Rating:** 4
**Confidence:** 4

**Summary:**

This paper presents WALT, a framework for constructing web “tools” from the functional elements of websites. The central idea is to enable an agent to broadly explore and interact with a website, identify its reusable functionalities (such as search, filtering, or posting), and abstract them into high-level tools that can be invoked directly in subsequent tasks. These tools allow the agent to operate at a more semantic level, avoiding fragile, step-by-step UI manipulations. Experiments on the VisualWebArena and WebArena benchmarks demonstrate that WALT achieves substantial improvements in both success rate and execution efficiency compared with previous web-agent baselines.

**Strengths:**

1. The core idea is novel and intuitive and abstraction is conceptually sound and potentially impactful.
2. The empirical results show consistent improvement over prior methods across two benchmarks.

**Weaknesses:**

1. The paper is difficult to follow, particularly in the method section. Many paragraphs are either overly verbose or lack essential details, making it challenging to reconstruct the full workflow. Logical connections between subsections are weak, and the role of each module is not clearly articulated. For example, the implementation and workflow of B_browser, which generates demonstrations, are insufficiently described.
2. The proposed approach requires significant offline effort to explore and build a tool set for each website. And these tools are effective only for one website. If the websites change, or when the agent encounters unseen sites, this method does not work.
3. While the offline stage costs a lot, the experiment section does not analyze the time cost, success rate, or resource consumption of tool construction.
4. The paper does not include experiments that directly evaluate the Tool Construction procedure itself (e.g., URL promotion, schema synthesis, validation loops). The current experiments merely show that “using tools” improves results but do not demonstrate that the proposed construction method is necessary or superior to simpler alternatives. There is no analysis of whether this complex construction pipeline is justified in terms of accuracy, efficiency, or cost.
5. The labels of Figure 4 overlap and the text is unreadable.

**Questions:**

1. If the system detects or constructs a very large number of tools, how does it manage them efficiently? Would the growing tool set cause degradation in planning or selection performance?
2. It remains unclear how this method was used for the benchmark experiments. Were the same websites used in both the construction and evaluation phases? If so, could there be potential leakage or overfitting to specific site structures?

---

> ### Author Response · Authors · 2025-11-20
> **Thank you for the thoughtful feedback**
>
> Thank you for recognizing WALT's novel and intuitive core idea. **Please see our General Response**, which presents new experiments on 139 real-world websites, detailed tool construction evaluation, and substantial presentation improvements that address your primary concerns about clarity, real-world generalizability, and construction cost analysis.
>
> **Presentation Clarity:**
> We have **completely restructured Section 3** to address your concerns around clarity, elaborated on the role of each component, and added a concrete example to ground the discussion.
>
> **Tool Management at Scale:**
> By design, WALT does **not** try to learn 100s of tools per website. Instead, we explicitly prompt discovery to design a "**minimal but flexible API**" (Figure 2, Section 3, Appendix prompt) that maximizes coverage while minimizing redundancy. This yields a very modest number of tools per website (~50 tools in total across the 6 domains in WebArena and VisualWebArena).
>
> At runtime, agents are only exposed to tools learned for that **specific** website. On VisualWebArena/WebArena, agents comfortably scale to 10-15 tools per site with **consistent performance gains**. Large action spaces may indeed hurt performance and are thus avoided by design.
>
> **Construction/Evaluation Split & Overfitting:**
> Tools are discovered on the same websites used for evaluation but in a completely **task-blind** manner. WALT reverse-engineers **site functionality** (search, filter, create) with no prior knowledge of future tasks: the agent simply sees a search feature, learns a `search_listings` tool, and ends up using it extensively for tasks ranging from "find cheapest kayak" to "find most recent listing" and "find boats in category X". While WALT may indeed "overfit" to website functionality (colloquially speaking - WALT is a training-free method), we follow best practices to ensure there is no "task" overfitting.
>
> Finally, we note that while learned tools are indeed website specific, they still generalize across: (1) **tasks** (same tool for diverse objectives), and (2) **models** (Table 2 ablations: GPT-4.1, Gemini-2.5-flash, GPT-5-mini all use the same tool set discovered with GPT-5, showing tools are not model-specific).
>
> **Tool Construction Evaluation:**
> Please see **General Response: "Tool Construction Details"**. Key numbers: 82.6% success rate (252/305 candidates), 1.75 validation attempts per tool, $1.67/tool cost. Figure 4(b) reports runtime success rates (e.g., `search_listings` 100% over 262 invocations).
>
> **Website Changes:**
> We acknowledge tool maintenance is challenging when websites change. Currently, **agentic fallback provides runtime resilience**—when tools fail, the agent executes on-the-fly. A naive maintenance strategy could be to trigger **tool relearning** if fallback frequency for a tool increases beyond a threshold. Automated on-the-fly patching remains future work (Section 5).
>
> We hope these revisions address your concerns and are happy to clarify any remaining questions.

---

> > ### Author Response · Authors · 2025-11-23
> >
> > We sincerely hope that our rebuttal has addressed the reviewer's concerns, and would be happy to answer any lingering questions. Otherwise, we would appreciate if they would consider updating their score in light of our response.

---

### Official Review · Reviewer_yMR2 · 2025-11-01

**Soundness:** 3
**Presentation:** 4
**Contribution:** 3
**Rating:** 6
**Confidence:** 3

**Summary:**

The paper proposes WALT, a novel framework that reverse-engineers website functionalities into deterministic, invocable tools. This abstract away frafile, step-by-step UI interactions, making browser automation more reliable. One core contribution is to replace brittle click-based sequences with robust URL manipulations through API reverseengineering. Evaluation was run on VisualWebArena and WebAreana. WALT achieves state-of-the-art success rates with fewer steps than the baselines.

**Strengths:**

- **Originality:** WALT is novel in that it reframing browser automation as demonstrate-generate-validate high-level tools corresponding to website-provided functionalities, which is intuitive and robust.
- **Significance:** WALT shows SOTA performance on the two evaluaiton benchmarks, with higher efficiency with a controlled baseline approach without tools [Figure 3].
- **Robustness and generalizability:** Tools span multiple categories (search, filter, content creation, communication) and remain reliable under diverse layouts.
- **Great qualitatitive analysis:** Detailed analysis and observation of composition, success rates, and action type of discovered tools [Section 4.5].

**Weaknesses:**

- **Cost not quantified.** The paper does not specify the cost of offline exploration/validation. Please report discovery time distributions and other costs per validated tool.
- **Generalizability / Practicality beyond benchmarks.** WALT's generalization to live, frequently changing websites (e.g., with CAPTCHA or A/B testing) remains untested; a small study on production sandboxes or WorkArena++ tasks would be better.
- **Presentation**
  - Citation format: Line 299

**Questions:**

1. Why does the ablation study focus on a single split (VisualWebArena Classifieds) but not on other splits or WebArena?
2. There is a large focus on promoting eligible UI chains to URL operations. What is the frequency of failure for URL promotion or schema inference? How does WALT handle this?
3. **Fairness of comparison.** Some baselines like Claude Computer-Use Agent may operate with different observation spaces/limits/paradigm. How do you ensure comparison with baselines is fair with no hidden advantages for WALT?

---

> ### Author Response · Authors · 2025-11-20
> **Thank you for the thoughtful feedback**
>
> Thank you for your thoughtful feedback and recognition of WALT's originality and robustness. **Please see our General Response**, which presents new experiments and detailed cost analysis that directly address your questions about tool construction costs, URL promotion frequency, and generalizability to real-world websites.
>
> **Tool Construction Costs & Generalizability:**
> Please see **General Response: "Tool Construction Details"** and **"Real-World Validation"** for comprehensive analysis. Highlights: 252 tools across 139 websites, $1.67/tool, break-even at ~14 tasks, 32% URL promotion rate, +20.5% SR on real-world sites.
>
> **Ablation Scope:**
> Classifieds was chosen for detailed analysis because it offers rich tool diversity (9 tools spanning discovery, content management, and communication) while remaining computationally tractable for multiple backbone ablations (GPT-4.1, Gemini-2.5-flash, GPT-5-mini). To demonstrate **cross-domain generalization**, we present **Online-Mind2Web ablations** (±tools, controlled baseline) across 139 real-world websites spanning e-commerce, healthcare, travel, education, and government -- showing consistent gains (+20.5% SR, 23.3% fewer steps). This validates that tool benefits generalize well beyond simulated benchmarks and specific domains.
>
> **Baseline Fairness (Claude Computer Use):**
> Thank you for raising this concern. For VisualWebArena, we benchmarked Claude Computer Use ourselves following best practices (full setup in Appendix Section A.4). We acknowledge the reported numbers are a **lower bound** due to API constraints: Bedrock API limits forced 1280×720 screenshot resizing, 30-step limits, and heuristic-based webpage detection that may fail in edge cases. These technical limitations make exact replication challenging.
>
> To address this valid concern, we compare **Online-Mind2Web results comparing against Claude's official leaderboard performance**. Here, WALT (51.2%) achieves near-parity with Claude's official 51.7% **without any specialized training for browser automation**. This demonstrates that tool discovery can rival model-level specialization in a fairer comparison setting. Our controlled ablation (WALT vs baseline, ±tools, same setup) isolates +20.5% contribution from tools alone.
>
> **Citation format:** Fixed (Line 299).
>
> We hope these revisions address your concerns and are happy to clarify any remaining questions.

---

> > ### Author Response · Authors · 2025-11-23
> >
> > We sincerely hope that our rebuttal has addressed the reviewer's concerns, and would be happy to answer any lingering questions. Otherwise, we would appreciate if they would consider updating their score in light of our response.

---

### Official Review · Reviewer_LBGY · 2025-11-05

**Soundness:** 3
**Presentation:** 3
**Contribution:** 3
**Rating:** 6
**Confidence:** 3

**Summary:**

The paper introduces WALT (Web Agents that Learn Tools), a new framework designed to make web agents more robust and efficient. It tackles a core problem: current web agents are "brittle" because they rely on step-by-step UI interactions (e.g., click, type) and heavy LLM reasoning for every single action. This approach easily fails when website layouts change or tasks become complex.

WALT’s solution is inspired by how humans browse the web. Instead of thinking about individual clicks, humans use high-level functions a website provides, like "search," "filter," or "sort". WALT is a framework that reverse-engineers a website's built-in functionality into a set of reusable, callable "tools".

For example, instead of an agent executing a long, fragile sequence of actions to find the "cheapest blue kayak," the WALT agent can invoke a single, robust tool it learned for that site, such as search(query='kayak', category='Boats', sort_by='price', order='asc'). This "abstracts away low-level execution" and shifts the agent's job from fragile, step-by-step reasoning to high-level planning and reliable tool invocation.

**Strengths:**

1.  It proposes a new paradigm for web automation that shifts from brittle, low-level UI sequences to robust, high-level tool invocation.

2. It introduces a "demonstrate-generate-validate" loop to autonomously create these tools.  A browser agent explores the site to demonstrate its functionality (e.g., using search with all its filters). A "tool generation agent" analyzes these traces to create a structured tool, prioritizing robust URL manipulation (API reverse-engineering) over simple UI replays. A test agent verifies that the newly created tool works correctly before it is registered for use.

3. WALT achieves significantly higher success rates on the Visual WebArena (52.9%) and WebArena (51%) benchmarks, outperforming prior methods.

4. By abstracting complex actions into single tool calls, WALT completes tasks using fewer steps and less LLM-dependent reasoning.

**Weaknesses:**

1. The paper frames WALT as a paradigm shift from “skills” to “tools,” but the distinction is not always clear. Prior work such as SkillWeaver (Zheng et al., 2025) and Hybrid Agent (Song et al., 2024) already explored higher-level abstractions (skills, APIs) that reduce reliance on brittle UI actions. WALT’s contribution risks being seen as a rebranding of “API induction” or “workflow abstraction” unless the conceptual boundary is sharpened.

2. While ablations are provided, they mostly show aggregate improvements (e.g., +2.6% from multimodal DOM parsing). The analysis does not deeply isolate why certain components matter or how they interact. For example, how much of the gain comes from URL promotion vs. schema validation vs. fallback strategies?

3. WALT excels at deterministic, schema-driven tasks (search, sort, CRUD operations) but struggles with compound reasoning tasks (e.g., “find the most expensive boat with an image showing it on water and then rate it”). These failures highlight that WALT’s abstraction layer may not handle tasks requiring joint optimization across structured and perceptual constraints.

**Questions:**

Evaluation is limited to WebArena and VisualWebArena, which are simulated benchmarks. While these are standard, they may not capture the full variability of real-world websites (CAPTCHAs, A/B testing, anti-bot measures, dynamic content). The paper acknowledges this but does not empirically test robustness outside controlled environments.

---

> ### Author Response · Authors · 2025-11-20
> **Thank you for the thoughtful feedback**
>
> Thank you for your positive assessment of WALT's novelty and empirical results. **Please see our General Response**, which presents new experiments on 139 real-world websites that directly address your primary concern about real-world robustness and generalization beyond simulated benchmarks.
>
> **Novelty vs Prior Work:**
> Please see **General Response: "Novelty: WALT vs Prior Skills and APIs"** for detailed comparison. Key distinction: WALT surfaces robust tools that encapsulate **website-provided functionality** rather than codifying existing agent behaviors by mining ad-hoc agent trajectories. This mirrors human web use—leveraging infrastructure web designers built, not patterns agents happened to fall into.
>
> **Ablation Depth & Real-World Robustness:**
> Please see **General Response: "Real-World Validation"** for Online-Mind2Web results (139 websites, +20.5% SR, 27 tool wins) and **"Tool Construction Details"** for construction success rates and composition breakdown.
>
> **Compound Reasoning:**
> We appreciate this observation. Upon deeper analysis, the example failure ("most expensive boat with image on water, then rate") stems from **VisualWebArena Classifieds' search implementation quirk**: searching for "boats" within the "boats" category yields different results than searching "boats" without a category filter—only one path returns the correct item. This execution-path dependency is an artifact of the synthetic benchmark rather than a fundamental WALT limitation. Real-world sites (e.g., Target, GameStop on Online-Mind2Web) don't exhibit such brittleness.
>
> More importantly, **Online-Mind2Web demonstrates successful compound reasoning** across multiple domains *e.g.* healthcare provider matching with filters (Figure 5), finding cheap travel deals with multiple constraints (Figure 8), and financial planning with multi-parameter calculators (Figure 5). These examples show WALT handling joint perceptual + structured constraints effectively. We acknowledge remaining limitations in Section 5.
>
> We hope these revisions address your concerns and are happy to clarify any remaining questions.

---

> > ### Author Response · Authors · 2025-11-23
> >
> > We sincerely hope that our rebuttal has addressed the reviewer's concerns, and would be happy to answer any lingering questions. Otherwise, we would appreciate if they would consider updating their score in light of our response.

---

### Author Response · Authors · 2025-11-20
**General response to all reviewers (1/2)**

We thank all reviewers for their thoughtful and constructive feedback. We appreciate the recognition of WALT's conceptual novelty (LBGY, FkiG) and strong empirical results (all reviewers). The primary concern across reviews is **generalizability to real-world websites beyond simulated benchmarks**. We address this directly with new experiments, along with clarifications on novelty, costs, and presentation improvements.

---

### Real-World Validation: Online-Mind2Web Benchmark

**We evaluated WALT on Online-Mind2Web**, a benchmark comprising **139 real-world websites** spanning e-commerce, healthcare, travel, education, and government domains (Target, GameStop, Healthgrades, Coursera, IRS, etc.). This directly addresses concerns about WALT's generalization to real-world websites, in the presence of CAPTCHAs, dynamic layouts, and production environments.

We first use WALT to discover tools on the 139 websites (2-3 per site to keep costs reasonable), and then provide these to the agent at runtime.

**Results breakdown:**
- **WALT learns useful tools**: WALT autonomously discovers **252** validated tools on Online-Mind2Web, and successfully completes (i.e. finishes without environment errors) 238/300 tasks. Compared to a controlled tool-free baseline, WALT:
   - **Improves success rate**: by **20.5\%** (42.9&rarr;51.2)
   - **Improves efficiency**: using **23.3\%** fewer steps on average (10.8&rarr;8.2).
- **27 tasks show "tool wins"**: cases where baseline failed but WALT used learned tools to succeed, spanning 24 different websites (examples in Fig. 5 and Appendix).
- **Real-world limitations** (CcSC, FkiG): 62 tasks fail either due to bot detection (35) or timeout errors (27). In total, 22 websites are *completely* untestable due to strong anti-bot measures (*e.g.*, Apartments.com, Cars.com, UPS.com), illustrating the messy reality of real-world web automation. Some of these may be circumvented with sophisticated stealth strategies but is outside the scope of our work.

**Complementarity of simulated and real benchmarks:**
Online-Mind2Web validates real-world generalization but is limited to **read-only tasks** (browsing, search), as authenticated operations would be unsafe. **WebArena/VisualWebArena enable the full task spectrum**: content management (create/edit/delete with authentication), communication (post, upvote), and complex authenticated workflows. WALT discovers bespoke tools for these operations (e.g., `create_listing`, `post_comment`) – functionality Online-Mind2Web cannot safely test. The benchmarks are complementary: simulated for task diversity, real-world for robustness.

**Comparison to Claude Computer Use (yMR2):** To address the reviewer's concern, we evaluate WALT offline on Online-Mind2Web and compare against Claude Computer Use's **official leaderboard results**: promisingly, we find that WALT achieves near-parity in performance (-0.5\% lower success rate, absolute), *without any specialized end-to-end training for computer use tasks*. This demonstrates that tool discovery can rival specialized model training—a more generalizable and cost-effective paradigm.

---

### Novelty: WALT vs Prior "Skills" and APIs

**Core insight:** Humans use **website-provided functionality** (search, filters, forms)—robust by design. Prior "skill" approaches solve an artificial problem: they induce ad-hoc patterns from agent behavior rather than leveraging this infrastructure.

**WALT's paradigm:** Build robust and efficient tools that exploit **website-provided** functionality.

|  | SkillWeaver / AWM / ASI | Hybrid Agent | **WALT** |
|--------|------------------------|--------------|----------|
| **Approach** | Agent-induced from successful trajectories | Curated API docs | **Systematic exploration of website functionality** |
| **Consequence** | Codify existing behavior | Reliant on human-written docs | **Reverse-engineer site infrastructure** |
| **Implementation** | Brittle UI action replay | API calls (when available) | **URL promotion + validated schemas + fallbacks** |
| **Validation** | Unit tests on synthetic inputs | N/A | **Stress-testing on pre-vetted inputs** |

**Key differences:**
1. WALT discovers what *websites provide*, not what *agents did* - mirroring human web use
2. No documentation required; autonomous reverse-engineering
3. Optimized for robustness via schema validation, selector stabilization, URL inference

These distinctions mark a paradigm shift from mining agent behavior to surfacing site functionality.

---

> ### Author Response · Authors · 2025-11-20
> **General response to all reviewers (2/2)**
>
> ### Tool Construction and Performance Details
>
>
> **Online-Mind2Web discovery phase (139 websites):**
> - **305 tool candidates attempted** across 139 websites (**~2.2 candidates per website**)
> - **252 tools successfully generated** (82.6% success rate, **~1.81 tools per website**, 92\% of websites with at least one tool)
> - **159 tools successfully optimized** (58.5% success rate)
> - **Average ~1.75 attempts until successful validation**
>
> **Tool composition (252 tools analyzed):**
>
> | Tool Type | Count | Percentage |
> |-----------|-------|------------|
> | URL Promotion | 80 | 31.7% |
> | UI Only (Deterministic) | 38 | 15.1% |
> | Agentic | 60 | 23.8% |
> | Mixed (UI + Agentic) | 74 | 29.4% |
>
> **URL promotion** (highly efficient) succeeds in ~32% of cases - a substantial fraction. The remaining 67% use deterministic UI or targeted agentic steps, both more robust than baseline step-by-step reasoning. Schema inference failures trigger validation feedback loops (avg 1.75 attempts per tool).
>
> **Tool Construction Cost & Economics (yMR2, CcSC):** We estimate the cost of tool generation on Online-Mind2Web, using GPT-5 pricing as an example: offline tool generation costs **$1.67 per tool** (*proposal* = $0.26, *demonstration* = $0.87, *generation* = $0.46, *testing* = $0.08). To put this in context, tool-free inference using the same models costs **$0.12 per task** – single tool generation thus "breaks-even" at only ~14 tasks on a website, after which every additional use is both faster (23.3% fewer steps) and more successful (+20.5% relative SR). This is without accounting for the fact that tools only need to be learned **once** and can be reused indefinitely on the same website.
>
> **Scalability (CcSC, FkiG)**: WALT **explicitly optimizes for minimal, composable tool sets** – we prompt the discovery agent to design a "minimal but flexible API" that maximizes coverage while minimizing redundancy (Section 3, Appendix system prompt). This design principle distinguishes WALT from prior work like SkillWeaver, which accumulates task-specific skills without strategic minimization. Online-Mind2Web demonstrates this works at scale: **252 tools across 139 websites** (~1.81 tools/site on average) with successful retrieval and execution.
>
> ---
>
> ### List of Paper Revisions (in blue)
>
> - **Section 3 (Approach)**: Completely restructured for clarity, added concrete example to ground discussion
> - **Section 4.5**: New subsection on Online-Mind2Web evaluation
> - **Section 4.6**: Additional analysis with cost analysis and qualitative examples
> - **Figure 4**: Updated to improve clarity
> - **Appendix**: Detailed comparison table to prior work, Additional qualitative examples on Online Mind2Web (Fig. 7 and 8)
>
> We believe these additions substantively address all major concerns. WALT demonstrates strong real-world performance, practical construction costs, and clear conceptual novelty.

---

### Meta-Review · Area_Chair_ssdk · 2026-01-11

**Summary:**

Reviewer LBGY (score 6)
had comments about (i) existing papers that argue for a similar schematization of the interactions, (ii) inadequate analysis of what part of the approach leads to improvements in the tasks, and (iii) performance on more complex task specifications.

Reviewer yMR2 (score 6)
wanted clarifications on the cost and generalizability to more realistic websites.

Reviewer CcSC (score 4)
had comments on the narrative being difficult to follow, generalizability of the tools discovered across websites, and wanted experiments that evaluate the procedure to construct the tools (as opposed to the usage of the constructed tools).

Reviewer FkiG (score 4)
made comments on scalability, long-term usage (because of the inability to address changes), mapping UI actions to URL actions (which is not true for many websites), and wanted more evaluation in real-world settings.

This papers demonstrates a framework for learning tools for web interaction tasks. The system, named WALT, takes advantage of website-specific APIs (search, filter, sort, comment etc.) to treat these as abstract actions that can be executed robustly instead of relying on brittle step-by-step actions in LLMs/VLMs. The reviews were somewhat brief, but the additional experiments provided by the authors during the rebuttal have significantly improved the paper. I recommend that this paper be accepted.

**Reviewer Concerns:**

Reviewer LBGY
The authors have argued that WALT (the approach in this paper) uses existing developer-designed schema rather than compressing interactions of a navigation agent into macro actions. That makes it different from existing work. But it is difficult to understand how such website provided functionality will be discovered. The response to complex reasoning tasks is also largely inadequate.

Reviewer yMR2
The rebuttal addressed questions regarding the cost, as also some clarification questions on ablation studies etc. The authors also provided new experiments of real-world evaluation (Online M2W dataset). The new results are quite impressive.

Reviewer CcSC
The authors rewrote the narrative in Section 3 which has improved the readability of the paper. WALT explicitly learns a few tools specific to each website, but which can be repurposed to do multiple tasks on that website. The rebuttal identified one limitation, namely inability to address changes in the websites.

Reviewer FkiG
The new experiments provided by the authors do a good job of addressing all comments. They also argue that the strong performance on Online M2W gives evidence of how WALT uses URL-based steps for 32% of the tasks and uses the UI/agentic steps for the remainder. The system can therefore take advantage of both kinds of schematization.

**Reviewer Scores:**

Reviewer LBGY
I think the score would remain the same, 6.

Reviewer yMR2
I think the score would continue to lean towards an accept, 6.

Reviewer CcSC
I think it is a 50-50 chance that the score would be 4 or 6.

Reviewer FkiG
I suspect the score would be update to a 6.

---

### Decision · Program_Chairs · 2026-01-26

Accept (Poster)